DATA RELEASE

# A draft genome assembly for the dart-poison frog *Phyllobates terribilis*

Roberto Márquez[1,*,†], Denis Jacob Machado[2,3,4,*,†], Reyhaneh Nouri[2,3], Kerry L. Gendreau[1], Daniel Janies[2,3], Ralph A. Saporito[5], Marcus R. Kronforst[6] and Taran Grant[4,*]

1 Department of Biological Sciences, Virginia Tech, Blacksburg, VA, USA
2 Center for Computational Intelligence to Predict Health and Environmental Risks, University of North Carolina at Charlotte, Charlotte, NC, USA
3 Department of Bioinformatics and Genomics, University of North Carolina at Charlotte, Charlotte, NC, USA
4 Laboratório de Anfíbios, Departamento de Zoologia, Instituto de Biociências, Universidade de São Paulo, São Paulo, Brazil
5 Department of Biology, John Carroll University, University Heights, OH, USA
6 Department of Ecology and Evolution, University of Chicago, Chicago, IL, USA

## ABSTRACT

Dendrobatid poison frogs have become well established as model systems in several fields of biology. Nevertheless, the development of molecular and genetic resources for these frogs has been hindered by their large, highly repetitive genomes, which have proven difficult to assemble. Here we present a draft assembly for *Phyllobates terribilis* (12.6 Gb), generated using a combination of sequencing platforms and bioinformatic approaches. Similar to other poison frog sequencing efforts, we recovered a highly fragmented assembly, likely due to the genome's large size and very high repeat content, which we estimated to be ≈88%. Despite the assembly's low contiguity, we were able to annotate multiple members of three gene sets of interest (voltage-gated sodium channels and *Notch* and *Wnt* signaling pathways), demonstrating the usefulness of our assembly to the amphibian research community.

**Subjects** Genetics and Genomics, Evolutionary Biology, Zoology

**Submitted:** 26 April 2024

\* Corresponding authors. E-mail:
rmarquezp@vt.edu;
dmachado@charlotte.edu;
taran.grant@ib.usp.br

† Co-first authors.

Preprint submitted at
https://doi.org/10.32942/X2ND11

## INTRODUCTION

The ability to generate genome assemblies is now available for virtually any organism from which DNA of a reasonable quality can be obtained. This has led to an explosion of sequencing efforts across the tree of life (e.g. [1–5]). In vertebrates, these efforts have led to significant comparative genomic coverage in some groups [3]. For instance, high-quality assemblies are available for at least one species in more than 92% of bird families [6]. In amphibians, however, this is not the case. Despite considerable efforts, as of April 2024 assemblies for only 121 species (1.4%) from 31 families (40%) were available in NCBI's database [7]. However, only 63 of them have N50 values ≥1 Kb, accounting for 0.7% of amphibian species (36% of families). The slow progress in amphibian genomics is due in large part to the technical challenges posed by the large, highly repetitive genomes of many species [7–10].

Among amphibians, the genomes of poison frogs in the family Dendrobatidae have been notoriously challenging to assemble, given their often large sizes (up to 13 Gb [11]), as well

as their high content and widespread distribution of repetitive elements [10, 12]. Over the past few decades, several features of this family's biology, such as a wide variety of parental care strategies [13], multiple cases of exuberant intraspecific variation in coloration (e.g., [14–16]), and the recurrent evolution of alkaloid-based chemical defense coupled with aposematic coloration [17–19], have garnered the attention of many ecologists and evolutionary biologists. However, the functional and molecular mechanisms underlying the majority of these phenotypes and their evolution remain largely unknown. The development of molecular tools for functional genetic and evolutionary studies in poison frogs has, nonetheless, been slow, in part due to the challenges involved in genome assembly. Currently, assemblies are only available for four closely-related species: *Oophaga pumilio* [12], *Oophaga sylvatica*, *Dendrobates tinctorius* [20], and *Ranitomeya imitator* [21]. An assembly for the more distantly related *Allobates femoralis* (family Aromobatidae) has recently become available.

Here, we present a draft assembly for a fifth dendrobatid species, *Phyllobates terribilis* (NCBI:txid111132), whose genome was recently estimated to be around 12 Gb in size [11]. Frogs in the genus *Phyllobates* are well known for secreting batrachotoxins (BTXs) [22, 23], a group of potent neurotoxins that target voltage-gated sodium channels [24–26], as well as their dynamically evolving warning-colorations [23, 27]. Further, the phylogenetic position of the genus *Phyllobates* as sister to a radiation of more than 50 chemically defended, aposematic species [19] makes this clade crucial to understanding the evolutionary origin of alkaloid-based chemical defense in dendrobatids. These features make *Phyllobates* frogs well-suited models for studies in multiple fields that can certainly benefit from improved genomic resources, such as neurotoxin resistance and physiology [28–30], chemical ecology [31–33], and evolutionary genetics [27, 34].

## MATERIALS AND METHODS

### Animal subjects

All animals used for tissue sampling were acquired from the pet trade (Josh's Frogs, Owosso, MI, USA) or from laboratory colonies maintained at the University of Chicago. Individuals were euthanized through either an overdose of topical benzocaine followed by pithing, or progressive cooling and flash-freezing in liquid nitrogen. Thigh muscle, tadpole tail muscle, or liver samples were then dissected in phosphate-buffered saline (PBS), and either stored at −80°C or immediately used for DNA extraction as detailed below. Frozen samples were processed within 6 months of collection.

### DNA sequencing

We combined multiple sequencing strategies to produce a hybrid genome assembly. First, we generated paired-end (PE) and mate-paired (MP) Illumina libraries, as detailed by Jacob Machado et al. [35]. Briefly, DNA was extracted from thigh muscle with Qiagen MagAttract HMW DNA extraction kits (cat. no. 67563), and PE and MP libraries were prepared using the TruSeq Nano DNA and Nextera Mate-pair kits, respectively. DNA for MP libraries was gel size-selected to insert sizes of 3, 5, 8, and 10 Kb prior to library preparation. All libraries were sequenced on HiSeq 2500 instruments (RRID:SCR_016383) with 125 bp or 150 bp reads.

Next, we used the Pacific Biosciences Sequel platform (RRID:SCR_017989) to generate long single-molecule real-time (SMRT) sequencing reads. High-molecular weight DNA was extracted from ~70 mg of tadpole tail muscle using a Qiagen DNeasy column, and eluted in

**Table 1.** Sequence data used in the *Phyllobates terribilis* assembly. Numbers correspond to raw data. Estimated coverage was calculated as the product of read length and number of reads divided by the estimated genome size (12.8 Gb [11]). PE: paired-end, MP: mate-paired, $\bar{x}$: mean, SD: standard deviation.

| Library | Read length | No. reads | Estimated coverage | SRA accession |
|---|---|---|---|---|
| Illumina PE | 150 bp | 263,828,699 | 6.28× | SRR27279919 |
| Illumina PE | 125 bp | 210,072,773 | 4.17× | SRR27279918 |
| **Total PE** | | **473,901,472** | **10.45×** | |
| Illumina MP - 3 Kb | 150 bp | 240,701,531 | 5.64× | SRR27279914 |
| Illumina MP - 3 Kb | 125 bp | 126,584,808 | 2.47× | SRR27279916 |
| Illumina MP - 5 Kb | 150 bp | 247,545,514 | 5.80× | SRR27279913 |
| Illumina MP - 5 Kb | 125 bp | 109,366,991 | 2.14× | SRR27279915 |
| Illumina MP - 8 Kb | 150 bp | 252,654,530 | 5.92× | SRR27279912 |
| Illumina MP - 10 Kb | 125 bp | 86,657,163 | 1.69× | SRR27279917 |
| **Total MP** | | **1,063,510,537** | **23.67×** | |
| Illumina *in-vivo* Hi-C | 100 bp | 919,642,651 | 14.37× | SRR27279911 |
| *P. bicolor* RNAseq | 100 bp | 256,624,922 | NA | SRR12232938 |
| PacBio Sequel | $\bar{x}$: 5,143 bp SD: 5,008 bp | 11,945,937 | 4.80× | SRR27279910 |

50 µl Qiagen AE buffer previously heated to 60°C. After quality control on an Agilent TapeStation (RRID:SCR_014994), one large-insert (10 kb) library was prepared and sequenced in 16 SMRT 1M cells at the Duke University Sequencing and Genomic Technologies Shared Resource.

Finally, we produced an *in-vivo* Hi-C library using the Proximo Hi-C Animal kit (Phase Genomics). The starting material was a mix of ~50 mg tail muscle and ~20 mg liver tissue from a single tadpole, and the manufacturer's protocol was followed with no modifications. The resulting library was size-selected for 300–700 bp fragments using SPRI magnetic beads (Sera-Mag), and sequenced on an Illumina HiSeq 4000 (RRID:SCR_016386) with 100 bp reads. All Illumina reads were quality-trimmed using Trimmomatic v. 0.39 (RRID:SCR_011848) [36], except for MaSuRCA v. 3.3.4 (RRID:SCR_010691) runs (see below). PacBio reads were converted to fasta format, filtered for sequences longer than 100 bp using BamTools (RRID:SCR_015987) [37], and error-corrected based on the Illumina PE reads with FMLRC [38]. Table 1 contains details and accession information for all sequence data used in this project.

## Assembly pipeline

As a first approach to generate a starting assembly, we used WTDBG v. 2.3 (RRID:SCR_017225) [39] with the PacBio reads as input. We used a k-mer size of 19 bp, and to account for our sequencing coverage, increased the k-mer sampling rate to 1/2 (-S 2) and retained contained reads for alignment (-A flag). This, however, resulted in a much smaller assembly than expected (1.96 Mb), likely due to the lower-than-ideal coverage of our PacBio data. We therefore used MaSuRCA v. 3.3.4 [40, 41] to produce a starting assembly from the Illumina PE and MP reads. Reads were not quality-trimmed, following developer guidelines [42]. The k-mer size for De Brujin graphs was determined automatically, and the coverage of MP libraries was limited to 300×. The MaSuRCA assembly was further scaffolded and gap-filled through a series of complementary approaches. First, we downloaded RNAseq data from a closely related species, *Phyllobates bicolor*, generated in a previous study (SRA accession SRX8741407, BioProject PRJNA645960 [27]). We quality trimmed these data as detailed above, and used P_RNA_scaffolder [43] for RNAseq-guided scaffolding (*P. terribilis* and *P. bicolor* shared a common ancestor roughly 2 million years ago [27]). We then

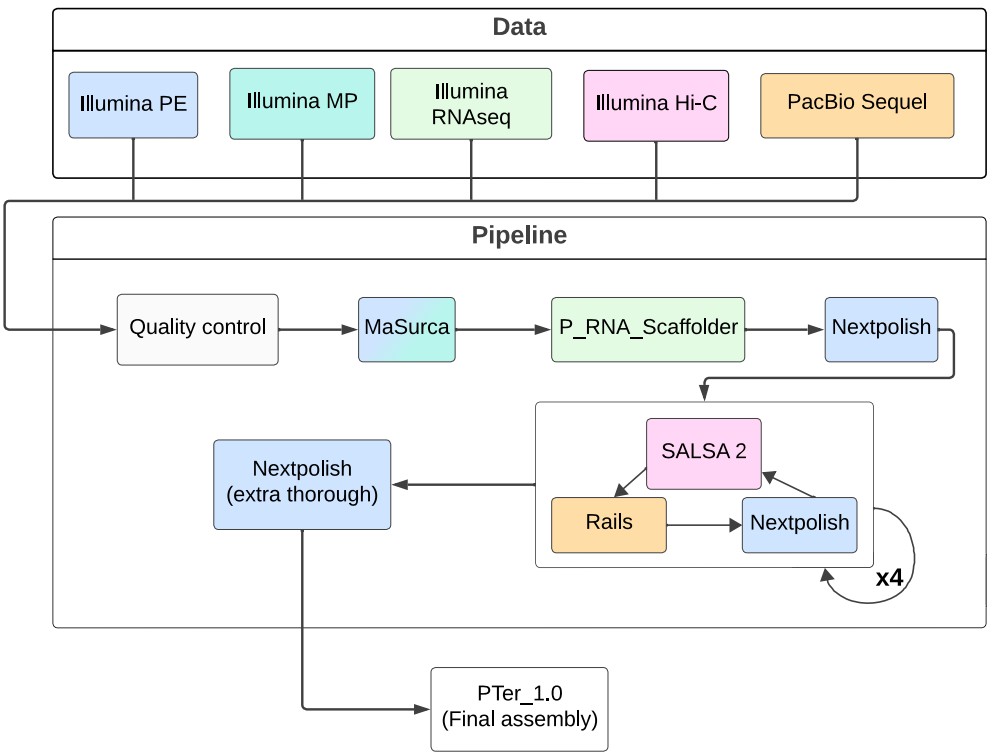

**Figure 1. Assembly pipeline.**
Pipeline used to generate a reference genome assembly for *Phyllobates terribilis*. Process boxes are filled with the same color as their input data.

polished the resulting assembly using the Illumina PE reads with Nextpolish (RRID:SCR_025232) [44], and ran four iterations of the following pipeline: HiC-based scaffolding with SALSA 2 (RRID:SCR_022013) [45, 46], PacBio-based gap-filling and scaffolding with LR_gapcloser (RRID:SCR_016194) [47], and RAILS [48], and polishing with NextPolish (RRID:SCR_025232; based on Illumina PE). At the end of the fourth iteration, we ran an additional three rounds of NextPolish. Read alignments for all steps were done with bwa v.0.7.17 (RRID:SCR_010910) [49], and alignment files were handled with samtools v. 1.11 (RRID:SCR_002105) [50]. Figure 1 summarizes our assembly pipeline.

Assembly contiguity statistics were calculated using quast v. 5.1 (RRID:SCR_001228) [51], base error rates (QV) were evaluated with GAEP [52], and completeness was assessed using BUSCO v. 5.3.2 (RRID:SCR_015008) [53, 54]. QV scores were calculated based on read mapping and called genotypes following Rhie *et al.* [4], using all available Illumina reads (except RNAseq). MP and HiC reads were not paired for mapping, and genotypes were called using bcftools (RRID:SCR_005227) [55]. BUSCO used the tetrapoda_odb10 gene set, and was run under default parameters. Upon upload to NCBI, contigs showing signs of possible contamination or shorter than 200 bp were removed.

To gain insight on the degree of missassembly due to repetitive content (e.g., multiple repeats being collapsed into a single sequence), we estimated copy numbers for regions annotated as repetitive by RepeatMasker (RRID:SCR_012954; details below) using DepthKopy [56], which uses sequencing depth at complete BUSCO genes as the expectation

for single-copy regions, to then estimate copy number at other regions of the assembly. In addition, we examined copy number variation in non-overlapping 5 Kb windows along the four longest scaffolds (17.4 Mb), of which 8.62 Mb (49.5%) were annotated as repetitive (see Repeat assembly and masking section below). Finally, to test for inflated coverage in repetitive regions, we compared the copy number of regions annotated as repetitive and single-copy BUSCO genes using negative binomial regression, as implemented in the R package MASS [57]. DepthKopy outputs copy number estimates on a continuous scale, so they were rounded to integers to match the discrete nature of the negative binomial distribution.

## Repeat assembly and masking

We used our assembly and raw PE reads to characterize the composition and distribution of repetitive elements in the *P. terribilis* genome. First, we used REPdenovo v. 0.1.0 [58, 59] to identify reads originating from repetitive elements, assemble consensus sequences for these repeats, and estimate their copy number based on read depth. Next, we used RepeatModeler v. 1.0.11 (RRID:SCR_015027) [60] to generate a species-specific repeat library for *P. terribilis*. We merged the repeat sequences from REPdenovo and RepeatModeler with those available in RepBase (2018 version; [61]) and the curated Dfam sequences distributed with RepeatMasker 4.0.8 [62], and used RepeatMasker to annotate and mask them on the assembly. The repeat-masked assembly was used in all subsequent annotation steps. Consensus repetitive sequences identified by REPdenovo were annotated by blasting [63] against the combined Dfam and RepBase nucleotide and protein databases ($E$-value $\leq 10^{-3}$), and retaining the hit with the highest bit score. The contribution of each consensus element to the genome-wide repetitive content was estimated by multiplying the length of the element by its mean coverage, and dividing this value by the total amount of sequence in the PE reads used as input. Finally, to assess how well the REPdenovo sequences were incorporated into our assembly, we queried them against the (unmasked) scaffolds using blastn (RRID:SCR_001598) ($E$-value $\leq 10^{-10}$).

## Gene prediction and annotation

We generated gene structure predictions using BRAKER v. 2.15 (RRID:SCR_018964) [64], based on a database of known proteins derived from UniProt's SwissProt and NCBI's RefSeq, and the *P. bicolor* RNAseq reads (see Table 1) aligned to the repeat-masked assembly. A transcriptome assembly generated previously from these reads [27] contained 80.8% of genes in the BUSCO tetrapoda_odb10 gene set (76.9% complete, 3.9% fragmented). The known protein database was generated by concatenating the SwissProt and RefSeq proteins for chordates (taxon code 7711), and removing duplicate or nested sequences, as well as those with duplicate headers, shorter than 32 amino acids, or marked as "partial" or "low quality". The filtered database consisted of 768,857 amino acid sequences. RNAseq reads were aligned to the assembly using STAR 2.7.9a (RRID:SCR_004463) [65], and BRAKER2 was run on a single core to avoid parallelization problems associated with fragmented assemblies. To evaluate the extent of gene representation and fragmentation in our assembly and annotation, we ran BUSCO on the resulting protein sequences as detailed above, and blasted the aforementioned *P. bicolor* transcriptome [27] against our assembly and annotation. Gene representation was assessed based on the proportion of transcripts



with blast hits ($E$-value $\leq 10^{-10}$), and completeness as the proportion of these transcripts that had query coverages of at least 75% (qcovs statistic from BLAST).

We then annotated the BRAKER2 gene sequence predictions by aligning and comparing them with multiple protein function and gene ontology (GO) databases. First, we used DIAMOND (RRID:SCR_016071) [66] to query gene predictions against the NCBI's non-redundant (NR) database, and InterProScan (RRID:SCR_005829) [67] to generate an initial prediction of protein function. We then used Blast2Go (RRID:SCR_005828) [68] to perform GO mapping and annotation based on the DIAMOND and InterProScan results, and extracted GO subsets (i.e., GO slims) based on the "generic GO subset" list available from the Gene Ontology Resource [69]. Gene predictions and annotations are available in the GigaDB repository associated with this paper.

## Targeted gene annotation

In addition to a general annotation of genes in our assembly, we evaluated its applicability through a systematic search for three gene sets of particular interest: The voltage-gated sodium channels, which have been frequently studied in the context of toxin resistance in poison frogs (e.g., [12, 29, 30, 70]) and other animals (e.g., [71–74]), and genes involved in the *Notch* and *Wnt* signaling pathways, which exhibit a high degree of conservation across the animal kingdom [75–78]. The *Notch* pathway plays a pivotal role in regulating a spectrum of fundamental cellular processes, including differentiation, fate specification, proliferation, programmed cell death, and tissue patterning, and has been extensively characterized across a variety of species at both embryonic and post-embryonic stages [76–82]. The *Notch* and *Wnt* pathways interact in diverse molecular, cellular, and developmental contexts, which have also received substantial attention [83–85]. A comprehensive enumeration of selected genes pertinent to the *Notch* and *Wnt* signaling pathways is presented in Figure 2.

Our goal here was to showcase how, regardless of its suboptimal state (see Results and discussion), our assembly can still be used to annotate genes of interest. We note, however, that our approach collapses closely related paralogous genes with conserved structure and function into a single gene name, since establishing orthology for these genes across species requires detailed annotation beyond the scope of this paper. For example, some gene names like "*Wnt*" represent a collection of paralogs, which vary in number and identity across species. With this in mind, the results of our targeted annotation must be interpreted bearing in mind that we cannot distinguish between finding some or all members of a closely related gene family. Moreover, it is important to emphasize that our targeted annotation approach was designed specifically to validate the presence of genes of interest rather than assess conventional metrics of genome completeness or gene model quality. Unlike traditional comparative analyses with well-annotated reference genomes (e.g., *Xenopus tropicalis*), our methodology focuses on gene presence validation through multiple independent methods. Many of the gene families we analyzed (such as voltage-gated sodium channels and Wnt signaling components) exhibit substantial variability in domain structure and sequence length, even among well-assembled genomes, making standardized length and completeness comparisons challenging. This inherent complexity in these gene families means that attempting to quantify completeness or provide consistent domain composition metrics would require extensive computational processing beyond the scope of this manuscript. This multi-method validation approach provides increased confidence in our annotations despite the fragmented nature of the assembly.



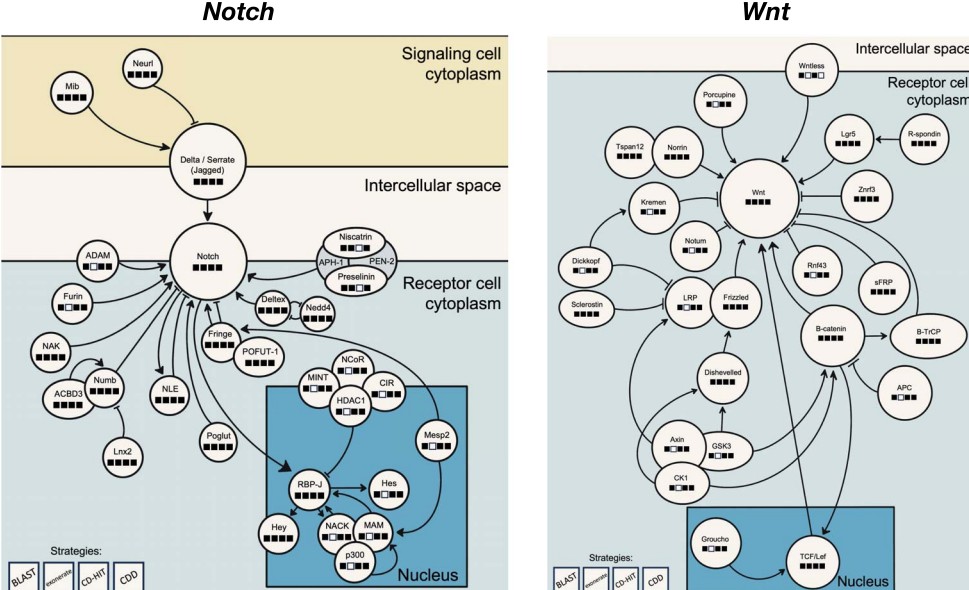

**Figure 2. Annotated pathways.**
Simplified diagrams of the *Notch* and *Wnt* pathways, whose genes we targeted for systematic annotation. Boxes below gene names indicate whether a protein product was found using each of the three different search strategies employed, and whether they fulfilled the conserved domain requirement (see Targeted gene annotation section for details). Some gene names, like "*Wnt*", group proteins with similar structures and functions encoded by different closely related genes. Filled boxes denote a positive result.

We employed three different sequence search algorithms to query a set of reference sequences for our genes of interest against the *P. terribilis* BRAKER2 gene predictions and assembly scaffolds, and then used NCBI's Conserved Domain Database (CDD; [86]) to identify conserved domains in the resulting hits in order to confirm that they represented actual matches to the target proteins. The reference database was generated from mouse, human, and amphibian sequences on RefSeq, UniProt, Xenbase, and previous publications [12], and was filtered as detailed above (see the Gene prediction and Annotation section). Initial sequence searches were conducted using tblastn (RRID:SCR_011822) [63], exonerate (RRID:SCR_016088) [87], and CD-HIT (RRID:SCR_007105) [88]. The resulting hits were then queried against the CDD using NCBI's CD-Search online portal [89] with an *E*-value cutoff of 0.01, and adjusting scores for sequence composition. Genes with conserved domains that departed from their putative function (i.e., *Notch* or *Wnt* signaling, or voltage-gated sodium channel) were discarded. The quality of a hit in our targeted annotation approach should be assessed by the number of independent sources confirming the presence of a gene (or gene family) in our assembly.

## RESULTS AND DISCUSSION

### Genome assembly and annotation

The final assembly, **PTer_1.0**, spans 4.24 Gb, and has a scaffold N50 of 11,957 bp, L50 of 93,411 fragments, and GC content of 42.26%. Scaffolds range in length from 63 to 5,200,876 bp. The final BRAKER2 annotation contains 45,969 protein-coding sequences with an average length of 211 amino acids (SD: 217.5 amino acids). BUSCO assessment found 40%

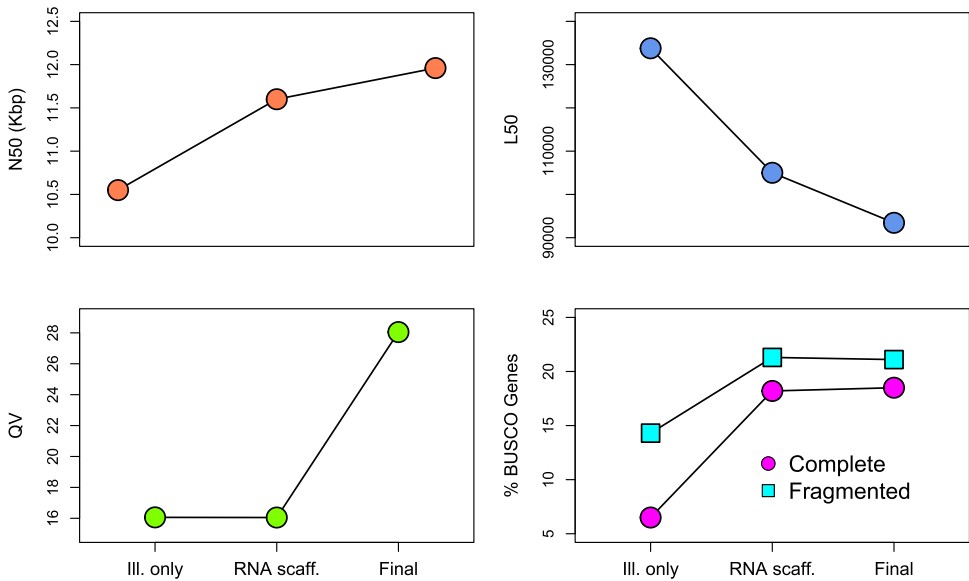

**Figure 3.** Contiguity and completeness statistics at three key steps of the assembly pipeline: the initial Illumina-only assembly generated with MaSuRCA, the assembly resulting from RNAseq-based scaffolding, and the final assembly incorporating PacBio and HiC scaffolding and gap filling. See Figure 1 in the main text for further details on the assembly pipeline, and the Methods section for details on each statistic.

**Table 2.** Contiguity and completeness statistics for the *P. terribilis* assembly and annotation.

|  | Length | N50 | L50 | QV | GC % |
|---|---|---|---|---|---|
| Contigs | 2.21 Mb | 1.06 kb | 601,531 | 28.05 | 42.26% |
| Scaffolds | 4.24 Mb | 11.96 kb | 93,411 | 28.02 | 42.26% |
| **BUSCO** | **Complete SC** | **Complete Dupl.** | **Fragmented** | **Missing** |  |
| Assembly - Unmasked | 18.5% | 0.4% | 21.1% | 60.0% |  |
| Assembly - Masked | 17.4% | 0.3% | 20.9% | 61.4% |  |
| Annotation | 14.0% | 0.3% | 23.4% | 61.3% |  |

of genes (18.9% complete, 21.1% fragmented) in the unmasked genome assembly and 38.7% of genes in the annotation (15.3% complete, 23.4% fragmented). Additional assembly statistics are available in Table 2, and statistics from intermediate steps in our pipeline are presented in Figure 3. The fragmentation in our assembly is probably due to a combination of factors. First of all, the highly repetitive nature of the *P. terribilis* genome is bound to generate assembly issues. Beyond this, its very large size means that our sequencing effort, despite being considerable, resulted in suboptimal depth, especially in terms of long-read data. Future assembly attempts should incorporate higher long-read depth, as well as longer reads that are able to transverse entire repetitive regions.

As with other poison frog assemblies based primarily on short-read data, the size of our assembled sequence is considerably smaller than genome size estimates based on read depth or DNA quantification. The current *Oophaga pumilio* reference assembly on GenBank (accession GCA_009801035.1 [12]) is 3.5 Gb, while the genome size for this species has been estimated at 4.3–4.7 Gb based on Feulgen staining [11, 90]. Our *P. terribilis* assembly is 4.2 Gb, while a genome size estimate based on BUSCO gene read depth (generated with DepthSizer (RRID:SCR_021232) [56]) was between 10.1 and 17.6 Gb, and the average DNA

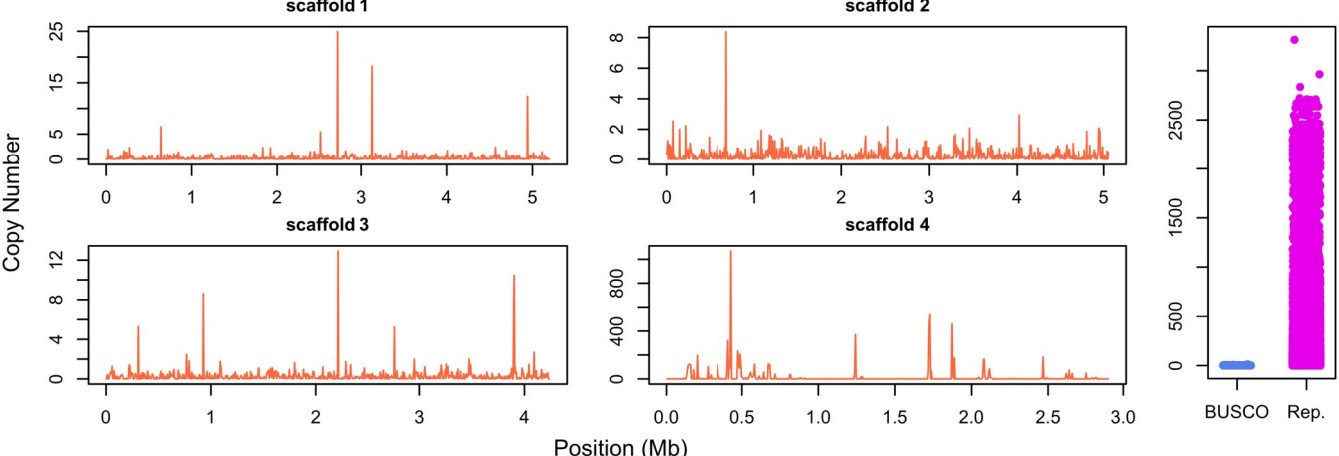

**Figure 4.  Copy Number Variation.**
Copy number variation across the four longest scaffolds of our assembly (left and center panels) and between silge-copy BUSCO genes and repetitive regions annotated by RepeatMasker (right panel), estimated using DepthKopy [56].

content in *P. terribilis* nuclei has been estimated at 12.88 pg using flow cytometry [11], which equates to 12.6 Gb. Despite this discordance, the vast majority of Illumina PE (96.3%) and PacBio (95.1%) reads successfully mapped to our assembly. However, we found pronounced variation in copy numbers across the four largest scaffolds, with some 5 Kb windows reaching values above 1,000. Repetitive regions annotated by RepeatMasker had significantly higher copy numbers than single-copy BUSCO genes (negative binomial regression: $\beta$ = 0.712, $z$ = 4.914, $p$ = 8.94 × 10$^{-7}$; Figure 4). In view of these results, and considering the high repetitive content of this and other dendrobatid genomes [10, 12, 20, 21], we consider it likely that the discrepancy between genome size and assembly size is due to multiple repetitive regions of the genome being collapsed into single sequences during assembly. This may also cause breaks in the assembly, which would explain the low contiguity. The fact that we also find a large discordance in repetitive element content and composition between the assembly and raw reads (see the Repetitive Content Section below) further supports these hypotheses, and highlights the challenge repetitive regions pose for dendrobatid frog genome assembly.

## Targeted gene annotation

Despite its fragmentation, our assembly contained most of the *P. bicolor* transcripts, and all the target genes (or gene families) in our three sets of interest. Ninety-nine percent of the *P. bicolor* transcripts had BLAST hits against the genome, and 96% against the predicted protein sequences from our annotation. However, only 25% (genome) and 4.4% (annotation) of these hits spanned at least 75% of the query transcript within the same scaffold/peptide. Similarly, all voltage-gated sodium channels and at least one member of the gene families involved in the *Wnt*/*Notch* pathways were represented in our assembly. Their sequences contained conserved domains concordant with their putative function, indicating that these sequences likely represent true members of their respective gene families (Figure 2). The strength of our annotation approach lies in the use of multiple independent validation methods (tblastn, exonerate, and CD-HIT, followed by conserved

**Table 3.** Composition of repetitive elements identified from the assembly by RepeatMasker and from PE reads by REPdenovo. RepeatMasker values are based on an assembly length of 4.24 Gb. Sequence lengths and percentages from REPdenovo correspond to values across the 65.83 Gb of PE reads used as input. Assuming the PE reads represent an unbiased sampling from the genome, percentages can be interpreted as estimates of the genome-wide repeat composition. Repeat contents and assembly sizes for two closely related species, *Allobates femoralis* and *Oophaga sylvatica*, are included, as reported in Table S4 by Kosch et al. [7].

| Class | Subclass | Repeat masker | | REPdenovo | | Other species (percents) | |
|---|---|---|---|---|---|---|---|
| | | Mb in genome | Percent | Gb in reads | Percent | *A. femoralis* (5.3 Gb) | *O. sylvatica* (5.2 Gb) |
| DNA | | 49.5 | 1.17% | 8.49 | 12.9% | 11.86% | 12.71% |
| | TcMar-Tc1 | 0.63 | 0.01% | 4.78 | 7.3% | 8.10% | 6.82% |
| | hAT-Ac | 0.18 | <0.01% | 3.38 | 5.1% | 2.80% | 4.91% |
| | Other | 48.7 | 1.16% | 0.33 | 0.5% | 0.96% | 0.98% |
| LTR | | 19.6 | 0.46% | 5.24 | 8.0% | 7.07% | 24.46% |
| | Gypsy | <0.01 | <0.01% | 4.35 | 6.6% | 5.82% | 21.82% |
| | Other | 19.6 | 0.46% | 0.89 | 1.4% | 1.25% | 2.64% |
| LINE | | 29.1 | 0.69% | 2.76 | 4.2% | 8.06% | 6.08% |
| | CR1 | 6.60 | 0.16% | 0.83 | 1.3% | 3.44% | 3.35% |
| | L1 | 2.84 | 0.07% | 0.96 | 1.5% | 2.87% | 0.51% |
| | Other | 19.66 | 0.46% | 0.97 | 1.5% | 1.75% | 2.73% |
| Other | | <0.01 | <0.01% | 0.62 | 0.95% | 1.94% | 2.99% |
| Unclassified | | 1,393.1 | 32.85% | 41.23 | 62.6% | 44.59% | 35.39% |
| **Total repetitive** | | 1,491.3 | 35.17% | 58.35 | 88.6% | 73.52% | 81.63% |

domain confirmation), which provides robust evidence for gene presence even in a fragmented assembly context. The identification of characteristic conserved domains confirms their identity and functional relevance.

Given the low completeness of transcript BLAST hits and the BUSCO statistics reported above, we anticipate that many of the gene families annotated are likely represented as fragments rather than complete sequences, and that at least some gene families will have missing genes. Without further detailed annotation, we cannot reliably evaluate how many unique orthologs within a gene family were annotated. Finally, the results above suggest that the low BUSCO scores obtained are at least partially due to fragmentation hindering BUSCO's ability to annotate genes, rather than the absence of at least partial or fragmented sequences of these genes in the assembly. In any case, these results highlight that, while suboptimal, our assembly remains a valuable resource for genetic research in dendrobatids and vertebrates in general.

## Repetitive content

Repetitive elements identified by RepeatMasker make up 1.49 Gb (35.17%) of the assembly, with the majority remaining unclassified after annotation (1.39 Gb; 32.9% of the assembly). REPdenovo, on the other hand, identified 58.35 Gb of repetitive sequence in our 65.83 Gb of PE reads (88.6%), suggesting a much higher repetitive content in the *P. terribilis* genome than what is currently in our assembly (Table 3). In addition to repetitive element collapse during assembly, this discordance is likely to be exacerbated by the high number of gaps in the assembly (Table 2), which may have precluded the discovery of repetitive elements by RepeatMasker. With this in mind, and considering that several assemblies of dendrobatid frogs are over 70% repetitive (Table 3 [10, 20, 21]), and that the *P. terribilis* genome is considerably larger than all other sequenced species [11], 88% is likely a closer estimate of the true repetitive content.

REPdenovo assembled the repeat-containing reads into 126,314 consensus sequences, of which 114,283 (90%) had BLAST hits against our assembly spanning at least 95% of the repeat with ≥98% identity. This, again, indicates that most repetitive elements were indeed

incorporated into the assembly, but were collapsed into a small subset of sequences due to their high degree of similarity. Blasting to the RepBase and Dfam protein libraries allowed us to annotate 22,952 of the REPdenovo repetitive elements, which together accounted for 17.12 Gb of the 58.35 Gb identified as repetitive. Among these, DNA, long terminal repeats (LTRs), and long interspersed nuclear elements (LINEs) were most prevalent, similar to other poison frog assemblies (Table 3 [12, 20, 21]).

Our finding that a considerable portion of repetitive elements was unclassifiable through comparisons with commonly used repeat databases has also been obtained by several other amphibian genome assembly efforts [10]. Although the fact that amphibian repetitive elements are not well represented in the databases used could in part explain this result, recent studies (e.g., [21, 91]), as well as our own ongoing work using recently generated amphibian repetitive element libraries, have found similar results, suggesting this may not be the case. Whether some unique feature of anuran repetitive elements, such as novel transposable element families, is behind the challenges with their classification remains to be determined. In-depth investigation of these unannotated sequences should, in any case, improve our ability to annotate and understand the evolution of repetitive elements in eukaryotic genomes.

## CONCLUDING REMARKS

With the rapid accumulation of genome assemblies for non-traditional model species, comparative genomics has gained ground as a powerful approach across the biological sciences. Our draft genome assembly for *P. terribilis* will contribute to research efforts in a variety of fields, including systematics, phenotypic evolution, chemical ecology, developmental biology, molecular physiology, and sensory ecology. Despite a considerable multi-platform sequencing effort, our assembly remains highly fragmented, likely due to its large size and the rampant proliferation of repetitive elements, which comprise as much as 88% of the genomic sequence. Although its current level of fragmentation is certainly an obstacle for some studies, such as those that rely on scoring features across long contiguous stretches of DNA (e.g., ancestry tracts or runs of homozygosity), our analyses suggest that, even in its current form, this assembly represents a valuable resource. As DNA sequencing technologies continue to improve and become more cost-efficient, we are confident that this work will constitute a key stepping stone towards chromosome-level assemblies for highly repetitive amphibian genomes.

## DATA AVAILABILITY

Our assembly and sequencing data are available under NCBI BioProject no. PRJNA1054463. Raw reads have SRA accessions SRR27279910–SRR27279919, as detailed in Table 1. The final assembly, PTer_1.0 has been assigned WGS accession JBBPXS01, and is available under GenBank accession GCA_045270155.1. The assembly (repeat-masked and unmasked), gene predictions and annotations, and REPdenovo and RepeatMasker output files, along with the code used to generate the assembly, are available in the GigaDB repository associated to this manuscript [92].

## LIST OF ABBREVIATIONS

BUSCO, Benchmarking Universal Single-Copy Orthologs; CDD, Conserved Domain Database; GO, gene ontology; LINE, long interspersed nuclear element; LTR, long terminal repeat; MP, mate-paired; PE, paired-end; SD, standard deviation; SMRT, single-molecule real-time.

## DECLARATIONS

### Ethics approval and consent to participate

All animal work was approved by the University of Chicago and John Carroll University's Institutional Animal Care and Use Committees (UChicago protocol #72416; JCU protocol #1400).

### Competing interests

The authors declare that they have no competing interests.

### Authors' contributions

RM, DJM, DJ, MRK, and TG designed the study. RM, DJM, and RAS performed laboratory work, RM, DJM, RN, and KLG conducted data processing and bioinformatic analyses, RM, DJM, DJ, MRK, and TG acquired funding. RM, DJM, and RN wrote the manuscript with input from all coauthors.

### Funding

This research was funded by the São Paulo Research Foundation (grant nos. 2012/10000-5, 2013/05958-8, 2015/18654-2, and 2018/15425-0), the Brazilian Conselho Nacional de Desenvolvimento Científico e Tecnológico (grant no. 314480/2021-8), the US National Science Foundation (grant nos. DEB-1702014 and IOS-1827333), the US National Institute of General Medical Sciences (grant no. R35GM131828), and the University of Chicago's Hinds Fund. Computations were supported in part by Advanced Research Computing at Virginia Tech, and the Center for Research Informatics at the University of Chicago. We acknowledge funding and logistical support from several entities of the University of North Carolina at Charlotte including: The Bioinformatics Services Division, the Department of Bioinformatics and Genomics, the Bioinformatics Research Center, University Research Computing, the College of Computing and Informatics. RM was supported by the Michigan Society of Fellows.

### Acknowledgements

We are grateful to Gabriel Massami Izumi de Freitas and Esdras Matheus Gomes da Silva for technical assistance, and to Carrie Olson-Manning and Adam Stuckert for advice with PacBio bioinformatics.

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
