## [Editor Report]

Editor’s AssessmentCompared to other vertebrate groups there is slow progress in amphibian genomics due in large part to the technical challenges posed by their large, highly repetitive genomes. Poison dart frogs (family Dendrobatidae) being a case in point with large (>10GB) genomes and high repeat content and widespread distribution of repetitive elements. This paper presents a draft assembly for the golden poison arrow frog Phyllobates terribilis, generating an assembly length of 4.24Gb for the predicted 12.6Gb genome using a combination of sequencing platforms (Hi-C, Illumina and PacBio Sequel) and bioinformatic approaches (eventually using MaSuRCA v. 3.3.4). The challenge assembling it fully being that as much as 88% of the genomic sequence was repetitive content. Peer review and additional polishing helped improve the assembly somewhat and demonstrate the data is usable. This work and approach this work hopefully constitute a key stepping stone towards chromosome-level assemblies for highly repetitive amphibian genomes. The genes annotated also providing a source of novel pharmacologically active batrachotoxins (BTXs), a group of potent neurotoxins that target voltage-gated sodium channels.Editor’s AssessmentCompared to other vertebrate groups there is slow progress in amphibian genomics due in large part to the technical challenges posed by their large, highly repetitive genomes. Poison dart frogs (family Dendrobatidae) being a case in point with large (>10GB) genomes and high repeat content and widespread distribution of repetitive elements. This paper presents a draft assembly for the golden poison arrow frog Phyllobates terribilis, generating an assembly length of 4.24Gb for the predicted 12.6Gb genome using a combination of sequencing platforms (Hi-C, Illumina and PacBio Sequel) and bioinformatic approaches (eventually using MaSuRCA v. 3.3.4). The challenge assembling it fully being that as much as 88% of the genomic sequence was repetitive content. Peer review and additional polishing helped improve the assembly somewhat and demonstrate the data is usable. This work and approach this work hopefully constitute a key stepping stone towards chromosome-level assemblies for highly repetitive amphibian genomes. The genes annotated also providing a source of novel pharmacologically active batrachotoxins (BTXs), a group of potent neurotoxins that target voltage-gated sodium channels.

---

## [Reviewer Report]

Indicate in the comments box below whether you are happy with the changes made or if the manuscript is unacceptable.Comments on revised manuscriptThe authors have engaged positively with the comments, and I think the clarity of the manuscript has been improved again. Whilst I would have liked to have seen more analysis of the targeted annotations - and feel that the usefulness of these data is limited without it - I appreciate that there are constraints on this kind of analysis. I am satisfied that the caveat regarding fragmentation is a sufficient compromise. The issues around the large genome size and relatively low sequencing coverage (<5X PacBio and ~34X Illumina in total) are now much clearer. Unfortunately, this has highlighted a concern that I raised in the first review and, on further reflection, was not adequately addressed, namely one of polishing and QV. It escaped my notice during the second review that the QV scores were so low. Can the authors please confirm that 13.89 is the correct Merqury QV score for the assembly, and clarify which reads Merqury used? I assume the complete set of Illumina reads was used for this, but it needs to be clearly stated in the methods. My major concern here is that if the details in the manuscript are correct, then (a) the assembly was polished seven times using only ~10X illumina data, and (b) the final QV score of the assembly indicates that the base accuracy of the assembly is much less than 99% (QV20), i.e. there is more than 1 error every 100 bp. This is considerably lower than raw Illumina sequencing data and is more akin to raw PacBio CCS, which is a major concern. (For contrast, the Earth Biogenome Project minimum standard is QV40, which is 1 error in every 10,000 bp.) I apologise for not picking up on this during the second review, but the assembly is not fit-for-purpose if this is a true reflection of the base accuracy. There are several possible contributing factors, and it is important that the authors identify and, preferably, rectify the problem. First, 10X is not really sufficient for polishing, especially with NextPolish. For example, this paper shows a signficiantly degraded performance under 30X: https://www.microbiologyresearch.org/content/journal/mgen/10.1099/mgen.0.001254. If the polishing only used the PE data, then it needs to be repeated using all of the available Illumina data. Second, NextPolish is designed to polish a long-read assembly, not a scaffolded short-read assembly, so perhaps its performance is suboptimal. I recommend also trying a more established and dedicated short-read polisher, Pilon, to see if it is more effective in this case. Third, seven rounds of polishing is a lot, especially with low coverage data. The major concern here is that polishing is introducing more errors than it is fixing. As requested in review 1, I would like to see how the QV changes with polishing - and know what the QV was for the original MaSuRCA assembly. This might be combining with the PacBio scaffolding strategy adding some very low-quality regions to the assembly. Finally, if Merqury itself was run with only 10X PE data (and not the full 34X PE+MP) it might be giving an inaccurate QV estimate. I recommend (1) re-running Merqury with all 34X Illumina - on at least the original MaSuRCA contigs and final assembly - and (2) re-polishing the genome (only once or twice) using Pilon and all 34X Illumina to see if the QV can be boosted to something respectable. Alternatively, if the authors spot a step in their assembly where the QV plummeted, they could consider reverting to that step. The potential good news is that this low QV is almost certainly contributing to the very low BUSCO scores, and so the contiguity/fragmentation might be slightly less of an issue than it currently appears. Either way, it would be useful to have all the assembly stats for the original MaSuRCA contigs in Table 2, including BUSCO. If the QV cannot be boosted, and the original MaSuRCA assembly was OK, the authors might even want to resort to using something like RagTag to scaffold the MaSuRCA assembly off their low-QV assembly and leave the PacBio data out of the final assembly.

---

## [Reviewer Report]

Indicate in the comments box below whether you are happy with the changes made or if the manuscript is unacceptable.Comments on revised manuscriptThe authors responded to my previous comments satisfactorily, however our colleague Dr. Rich Edwards brought up good specific criticisms that makes me want to address two points more specifically. (1) The authors claim in the Repetitive Content section that the closest identified repeats are from Xenopus, which is diverged ~200 million years. This is untrue as of earlier last year, when Bredeson et al 2024 did a denovo repeat annotation in both pipids and neobatrachians. The authors should use this database of repeats which is more complete for long read amphibian assemblies as is. Also, RepeatModeler is not mentioned in the Repetitive Content section, but it is standard practice to run on a species like this when no closely related group's repeats could be identified. What is the reason for not following the standard repeat annotation pipeline? If the computational resources necessary are too much, that is understandable and an acceptable reason, however it should be stated. (2) I think Dr. Edwards has a very good point in comparing the annotation quality of the specific genes that were focused on. If those genes are of large interest, we would want to be able to understand how fragmented these genes are compared to a well annotated species like Xenopus tropicalis. Overall if these comments are addressed I'd support publication to make this data available.

---

## [Reviewer Report]

Indicate in the comments box below whether you are happy with the changes made or if the manuscript is unacceptable.Comments on revised manuscriptOverall, I think the authors have made a good effort at addressing the reviewers’ comments and the manuscript is clearer as a result. However, a few more minor changes and additions are still recommended and, in some cases, required. My main comment is that I think the massive genome size for this species needs to be stressed more, as it is likely to explain a lot of the poor statistics. It should get a mention in the abstract and the estimated genome size from flow cytometry should be mentioned in the Introduction so the 12.8 Gb figure for Merqury is clear. Table 1 should be updated to included read volumes (Gbp) and estimated depth of coverage (X) for each technology. I would also like to see the totals across all technologies. Similarly, the (filtered) volume and coverage of reads should be presented in the assembly section. The PacBio read length is also pretty short, which will have contributed to the problems. Was DNA extraction/quality a problem? The choice for using 10 kb inserts needs some kind of explanation. It would be good to have some speculation based on all this hard work as to where the best future gains would be - it is quantity or quality of sequencing data that is the biggest hindrance for this large, repetitive genome? I suspect the read length distribution is a bigger factor for contiguity here than percentage repeats. Even though I suspect it would not work very well, it would be good to add GenomeScope kmer predictions of genome size too. Whilst not essential, I think the manuscript would benefit from a bit more discussion/analysis of the unique genome size and the repetitive genome size in Gb, as well as the percentage repeats. In Table 3, it would be good to include some stats from other closely-related species (including genome size) for context. Whilst the genome is clearly massively under-represented overall, it is not clear how much of the unique genome content is missing or collapsed. Related to this and the large genome size, what is the complexity of the Hi-C library? Does this account for the poor scaffolding? Would scaffolding the repeat-masked sequences have more success? I would also like to see a bit more discussion and analysis of the annotation, as the authors are proposing this (particularly for specific genes) as one of the useful aspects of the assembly. The assembled transcriptome is a lot more complete than the assembly and should be used to provide additional context. In particular, it would be useful to know if the low BUSCO completeness is due to fragmentation or missing sequences, and how badly affected the annotation is by this fragmentation. An average protein length of 211 aa is very short. How many of the transcripts (a) map to the assembly, and (b) map full length? Is it possible to pull out a subset of high-quality annotations that appear to be full-length and could be useful for phylogenomics etc? (The cane toad draft genome paper provides an example of this kind of analysis.) Related to the above, the details of all the identified genes in Fig 2 need to be provided as supplementary information. The concern here is that there are fragments and not whole genes. The lengths and domain composition for each protein should be provided, along with some exemplars from model organisms for comparison. Minor correction: DepthSizer bases its genome size prediction on BUSCO gene read depth, not coverage.

---

## [Reviewer Report]

Reviewer name and names of any other individual's who aided in reviewer Adam SessionDo you understand and agree to our policy of having open and named reviews, and having your review included with the published papers. (If no, please inform the editor that you cannot review this manuscript.)YesIs the language of sufficient quality?YesPlease add additional comments on language quality to clarify if needed
Are all data available and do they match the descriptions in the paper? YesAdditional CommentsAre the data and metadata consistent with relevant minimum information or reporting standards? See GigaDB checklists for examples <a href="http://gigadb.org/site/guide" target="_blank">http://gigadb.org/site/guide</a>YesAdditional CommentsIs the data acquisition clear, complete and methodologically sound?YesAdditional CommentsOverall the methods are sound however I am unsure how the genome quality is so bad given the data. Even in other more highly repetitive frogs, such as Rana muscosa, this combination of data yielded a chromosome-arm level assembly prior to manual analyses bringing it to chromosome scale.Is there sufficient detail in the methods and data-processing steps to allow reproduction?YesAdditional CommentsIs there sufficient data validation and statistical analyses of data quality? NoAdditional CommentsNot much validation of the assembly methods that I think may not be ideal for this genome.Is the validation suitable for this type of data?NoAdditional CommentsIs there sufficient information for others to reuse this dataset or integrate it with other data?YesAdditional CommentsAny Additional Overall Comments to the AuthorRecommendationReject (Unsound or Unusuable)

---

## [Reviewer Report]

Reviewer name and names of any other individual's who aided in reviewer Richard EdwardsDo you understand and agree to our policy of having open and named reviews, and having your review included with the published papers. (If no, please inform the editor that you cannot review this manuscript.)YesIs the language of sufficient quality?YesPlease add additional comments on language quality to clarify if needed
Are all data available and do they match the descriptions in the paper? NoAdditional CommentsThe descriptions in the paper appear to be correct. However, the genome and GigaDB data are not yet available.Are the data and metadata consistent with relevant minimum information or reporting standards? See GigaDB checklists for examples <a href="http://gigadb.org/site/guide" target="_blank">http://gigadb.org/site/guide</a>YesAdditional CommentsCannot check the genome itself. Whilst not required, it is recommended that the authors get a ToLID for the individual used for the genome sequencing.Is the data acquisition clear, complete and methodologically sound?YesAdditional CommentsThe choice of sequencing technology suggests that this was a study conducted over several years and updated as newer technology became available. It would be useful to clarify how long samples and/or DNA were stored prior to library preparation and sequencing. Likewise, the decision to use short reads for the core assembly and long reads for gap-filling and polishing would benefit from additional justification. Whilst not definitely wrong, the normal approach with that mix of technologies would be to perform a long-read assembly with additional mate-pair scaffolding and paired-end polishing to correct indels. Perhaps there was insufficient depth of sequencing for this? Table 1 should include volumes and estimated depths of sequencing, as well as no. reads. (If I understand correctly, they have about 61.4 Gbp CLR reads, which could be as low as 4.8X for a 12.8 Gbp genome.) Did the authors try converting their Sequel data to CCS reads to increase read accuracy? With a relatively low mean length of 5kb, I would imagine that the majority of ZMWs would produce CCS reads. If this and/or a long-read assembly was tried and failed to produce an improvement, this would be useful information to provide, to save future wasted effort.Is there sufficient detail in the methods and data-processing steps to allow reproduction?NoAdditional CommentsInsufficient program settings have been provided. Were all programs ran with default settings unless otherwise specified? The NextPolish polishing is not clear. Did this just use the (low coverage?) PacBio CLR reads?Is there sufficient data validation and statistical analyses of data quality? NoAdditional CommentsThe final assembly statistics are not clear. Standard stats, including contig and full BUSCO completeness stats, should be presented in a table. Merqury, or something similar, should be used to estimate the base-accuracy "QV" (or error rate) for the assembly. Ideally, this should be performed along with completeness estimates for each round of gap-filling/polishing. Note that gap-filling may cause the accuracy to go down by introducing more hard-to-assemble regions, so the QV should be calculated before and after each round of polishing. Seven rounds of polishing is a lot, especially if low-depth CLR reads are being used. Over-polishing can introduce errors. Were there any quality checks performed at each stage? A BUSCO completeness of 22.9% is incredibly low and signifies a major problem. Was the original raw assembly this low? Is this a problem of the fragmentation or low contiguity, or is it caused by over-polishing? What was the contig N50? Is it worse than the raw PacBio N50? This calls the assembly strategy into question (see above). Or was it an issue with the repeat masking? What are the BUSCO stats for the unmasked assembly? How much scaffolding was P_RNA_Scaffolder and the HiC able to do? Did the authors try a different program, such as YAHS or 3d-dna? (DNA Zoo assembly a lot of short-read assemblies to chromosome-level.) What was the complexity of the HiC library? At the very least, the number and N50 for scaffolds should be reported for the raw assembly and each round of scaffolding. There appears to have been no purging of redundant or low-quality sequences following gap-filling. Is this correct? I think it would be worthwhile running purge_dups, Diploidocus, or something similar to see whether any of gap-filling makes excess contigs redundant. This could be adding to the problems of the scaffolders. A mean protein length of 211 aa seems short, which is not surprising if the annotation is fragmented due to low contiguity. BUSCO should be performed on the annotation. It is probably a good idea to try a Trinity assembly of the RNASeq data and use this to help assess the quality of the assembly and its annotation. The quality of the targeted gene annotation is unclear. There should a Supplementary Table of all these predicted genes and their statistics (e.g. length, domain composition, and percentage coverage of the closest reference). Given the poor stats for the main annotation, I would find it very surprising if this represents a complete set of full-length genes. (Why were these genes singled out as being of interest?)Is the validation suitable for this type of data?NoAdditional CommentsPlease see requested additional validation above.Is there sufficient information for others to reuse this dataset or integrate it with other data?NoAdditional CommentsThere are two major limitations at present. The first is a good understanding of the actual quality of the assembly (see above). The second is a good understanding of how collapsed the assembly is. The authors have made some efforts in this latter respect, but the conclusions were a little unclear and some additional analysis would help. Secondly, it would be good to get a better handle on the genome size? Extrapolating from repeat content is a good idea, but the current data is hard to interpret. If the raw reads are 88.6% repeat and the assembly only 35.7%, that would imply that the 2.75 Gbp non-repeat assembly is actually 11.4%, which would extrapolate to 24.1 Gbp! This would suggest that the two definitions of "repeat" are not comparable - Table 2 should be extended to include the same classifications of the RepeatMasker runs using the REPdenovo repeats. A depth-based size predictor, such as DepthSizer, would probably give a more reliable estimate than kmer/repeat approaches, given the repeat content, and work out what the actual read depth should be. Finally, whilst Fig 3 gives some nice hints about repeat collapse, it is not very clear and represents <0.5% of the assembly. Would it be possible to summarise the whole genome? A program like DepthKopy, for example, can produce depth distribution summaries for scaffolds, windows and/or annotation.Any Additional Overall Comments to the AuthorIn the interests of full disclosure: Diploidocus, DepthSizer and DepthKopy are both my own programs. It is the principle that is important rather than use of those specific tools.RecommendationMajor Revision